# Selective RET Inhibitors (SRIs) in Cancer: A Journey from Multi-Kinase Inhibitors to the Next Generation of SRIs

**DOI:** 10.3390/cancers16010031

**Published:** 2023-12-20

**Authors:** Liz Clark, Geoff Fisher, Sue Brook, Sital Patel, Hendrik-Tobias Arkenau

**Affiliations:** Ellipses Pharma, London W1J 8LG, UK; liz@ellipses.life (L.C.); geoff@ellipses.life (G.F.); sue@ellipses.life (S.B.); sital@ellipses.life (S.P.)

**Keywords:** next-generation selective RET inhibitor (SRI), non-small-cell lung cancer (NSCLC), thyroid cancer, *RET* mutations, *RET* fusions, selpercatinib, pralsetinib

## Abstract

**Simple Summary:**

Since the discovery of the *RET* gene in the early 1980s, multiple treatments have been developed that can inhibit abnormal RET signaling. The first treatments were repurposed multikinase inhibitors, however, their low selectivity for RET led to unacceptable off-target toxicities and sub-optimal exposure in patients. More RET-specific or selective RET inhibitors (SRIs; selpercatinib and pralsetinib) were developed and were subsequently approved and are now established as standard of care treatment across a variety of indications including lung and thyroid cancer. However, there is now a need to develop treatment strategies that can address acquired resistance to these agents, including the development of next-generation SRIs and novel combination approaches. There also remains an opportunity to improve on the efficacy and safety/tolerability profile of the currently approved SRIs, especially to support potential combination approaches. This review discusses the current treatment landscape for RET-altered tumors and progress in the development of the next generation of SRIs.

**Abstract:**

RET is a receptor tyrosine kinase that plays an important role in the development of neurons and kidneys. The gene encoding the rearranged-during-transfection (*RET*) receptor tyrosine kinase was first discovered in the 1980s. Activating *RET* mutations and rearrangements have since been identified as actionable drivers of oncogenesis in numerous cancer types and are most prevalent in thyroid and non-small-cell lung cancer. Following the modest success of repurposed RET-active multikinase inhibitors, the first selective RET inhibitors (SRIs), selpercatinib and pralsetinib, received regulatory approval in 2020. Now, thousands of patients with RET-altered cancers have benefited from first-generation SRIs, with impressive deep and durable responses. However, following prolonged treatment with these SRIs, a number of acquired on-target resistance mutations have been identified together with other non-RET-dependent resistance mechanisms. Today, the focus is on how we can further evolve and improve the treatment of RET-altered tumors with next-generation SRIs, and a number of candidate drugs are in development. The ideal next-generation SRIs will be active against on-target acquired resistance alterations, including those that emerge in the CNS, and will have improved safety and tolerability relative to first-generation SRIs. In this review, we will provide an update on these candidates and their potential to meet the unmet clinical need for patients who progress on first-generation SRIs.

## 1. Introduction

In recent years, numerous molecularly targeted anti-cancer drugs have been developed based on a growing understanding of the role of proto-oncogene alterations in cancer pathophysiology. Indeed, molecularly targeted agents have become a standard-of-care treatment approach in many cancer settings. The *RET* (rearranged during transfection) proto-oncogene encodes a transmembrane receptor tyrosine kinase that is involved in normal embryonic development of neurons and kidneys [1]. RET can be activated through two main mechanisms: firstly, sporadic or germline *RET* mutations may directly or indirectly lead to activation of the kinase domain; and secondly, chromosomal rearrangement can result in hybrid proteins that fuse the RET kinase domain with a partner protein containing the dimerization domain. Both cases lead to RET-ligand-independent, constitutively active downstream signaling through pathways including RAF/MEK/ERK and PI3K/AKT, resulting in increased cell survival, invasion, and angiogenesis [2].

## 2. Oncogenic *RET* Alterations and *RET* Fusions

### 2.1. Non-Small-Cell Lung Cancer (NSCLC)

In 2020, lung cancer was the second-most common diagnosed cancer worldwide, with an estimated 2.2 million cases, and the leading cause of cancer death, with an estimated 1.8 million deaths (18% of total cancer deaths) [3]. NSCLC is the most common type of lung cancer, accounting for 81% of all lung cancer diagnoses [4]. The incidence of *RET* fusions is approximately 1–2% in NSCLC [2], and *RET* fusions appear to be associated with a high risk of brain metastases [5].

The baseline tumor characteristics and treatment responses to systemic chemo-immunotherapy from a large cohort of patients with *RET* fusion-positive NSCLC, untreated with SRIs, have recently been reported [6]. The median age was 63 years, 56% were female, and approximately 40% were smokers. Tumors were characterized by a low tumor mutational burden and low PDL1 expression. The most frequent fusion type was *KIF5B*, occurring in 72% of cases. In terms of metastases, the most common sites were lung (50%), bone (43%) and pleura (40%), with CNS disease found in approximately 25% at diagnosis of advanced disease. The response rates (55% and 46%) and median PFS (8.7 and 9.6 months) were similar for platinum-doublet chemotherapy alone or in combination with a checkpoint inhibitor. The median overall survival was approximately 16 months.

### 2.2. Medullary Thyroid Cancer (MTC)

The incidence of thyroid cancer has increased over the past decade, with MTC representing 1–5% of all thyroid cancer cases (75% sporadic and 25% hereditary). Despite its low prevalence, MTC accounts for almost 14% of all thyroid cancer-related deaths [7].

MTC originates from parafollicular C cells and can be hereditary. It is associated with two subtypes of multiple endocrine neoplasia syndrome type 2 (MEN2; MEN2A and MEN2B) or is sporadic. *RET* mutations occur in more than 95% of hereditary and approximately 50% of sporadic MTCs [7,8].

### 2.3. Other RET-Altered Solid Tumors

Papillary thyroid cancer accounts for 80–85% of all differentiated thyroid cancer. *RET* fusions have been described in patients with papillary thyroid carcinoma, accounting for approximately 20–40% of sporadic cases, with a higher frequency of *RET* fusions observed after radioiodine exposure [9]. *RET* alterations are also reported in a variety of other solid tumors with a background prevalence of approximately 1–2% and include colorectal, breast, paraganglioma, and urothelial carcinoma amongst others [9].

### 2.4. Treatment of RET-Altered Tumors—A Changing Paradigm

Multi-targeted kinase inhibitors (MKIs) such as cabozantinib, vandetanib and lenvatinib, initially developed to treat a number of clinical indications, have been used off-label to treat *RET*-altered tumors on the basis of modest response rates. For example, cabozantinib was studied in *RET*-rearranged lung cancers [10] and showed an overall response rate (ORR) of 28% (95% CI: 12–49), with progression free survival (PFS) of 5.5 months (95% CI: 3.8–8.4). Lenvatinib was studied in patients with *RET* fusion-positive NSCLC [11], with an ORR of 16% (95% CI: 4.5–36.1) and PFS of 7.3 months (95% CI: 3.6–10.2).

Clinically, management of side effects with MKIs, such as hypertension, proteinuria, GI toxicities, and bleeding, amongst others, made long-term dosing difficult, leading to dose reductions and discontinuation of treatment. In addition, treatment with MKIs were associated with emergence of mutations in the *RET* V804 gatekeeper residue, leading to acquired resistance and tumor escape [12].

### 2.5. First-Generation Selective RET Inhibitors

Selpercatinib (Loxo-292), developed by Loxo Oncology, Inc., Stamford, CT, USA, and Eli Lilly and Company, Indianapolis, IN, USA, inhibits wild-type *RET* and multiple mutant or fused *RET* subtypes (i.e., *KIF5B-RET* fusion protein −/+ the RET^V804M^ gatekeeper resistance mutation or the common *RET*-activating mutation M918T) [13,14].

In the LIBRETTO-001 open-label phase I/II trial, selpercatinib was administered orally to a total of 531 patients (dose escalation of 20–240 mg once or twice daily; patients in phase II received 160 mg twice daily), continuously in 28-day cycles, [15,16]. There were 144 NSCLC patients, 143 MTC patients, and 19 *RET* fusion–positive thyroid cancer patients evaluable for efficacy.

In the first 105 patients with *RET* fusion–positive NSCLC, with prior platinum-based chemotherapy, the ORR was 64% (95% CI: 54–73). The median duration of response was 17.5 months, and 63% of the responses were ongoing at a median follow-up of 12.1 months. In 39 previously untreated patients, the ORR was 85% (95% CI: 70–94), and 90% of the responses were ongoing at 6 months. Among 11 patients with measurable CNS metastasis at enrollment, the percentage with an objective intracranial response was 91% (95% CI: 59–100). The most common adverse events (AEs) of grade 3 or higher were hypertension (14% of patients), increased alanine aminotransferase level (ALT, 12%), increased aspartate aminotransferase level (AST, 10%), hyponatremia (6%), and lymphopenia (6%).

In the first 55 consecutively enrolled patients with *RET*-mutant MTC who had previously received vandetanib, cabozantinib, or both, 69% had a response (95% CI: 55–81) and 1-year PFS was 82% (95% CI: 69–90). In 88 patients with *RET*-mutant MTC who had not previously received vandetanib or cabozantinib, 73% had a response (95% CI: 62–82) and 1-year PFS was 92% (95% CI: 82–97). In 19 patients with previously treated *RET* fusion–positive thyroid cancer, 79% had a response (95% CI: 54–94) and 1-year PFS was 64% (95% CI: 37–82). The most common AEs of grade 3 or higher were hypertension (21% patients), increased ALT (11%), increased AST (9%), hyponatremia (8%), and diarrhea (6%).

A total of 12 of 531 patients (2%) discontinued selpercatinib because of a drug-related adverse event. Overall, selpercatinib was well tolerated with durable efficacy, and subsequently received accelerated approval based on these initial data from the first 144 NSCLC patients, 143 MTC patients and 19 *RET* fusion–positive thyroid cancer patients in the LIBRETTO-001 trial [15,16].

Selpercatinib was the first SRI to be approved by the FDA (8 May 2020) and by the EMA (10 December 2020). At that time, selpercatinib was indicated for the treatment of metastatic *RET* fusion–positive NSCLC, advanced or metastatic *RET*-mutant MTC (in patients who require systemic therapy), and advanced or metastatic *RET* fusion–positive thyroid cancer (in patients who require systemic therapy and are radioactive iodine-refractory (if radioactive iodine is appropriate) [17].

Following the review of updated results from an additional 172 patients and 18 months of additional follow-up to assess durability of response, the FDA subsequently granted regular approval to selpercatinib (21 September 2022) for adult patients with locally advanced or metastatic NSCLC with a *RET* fusion, as detected by an FDA-approved test. At the same time, the FDA granted accelerated approval to selpercatinib for adult patients with locally advanced or metastatic solid tumors with a *RET* gene fusion that have progressed on or following prior systemic treatment, or who have no satisfactory alternative treatment options. This was the first tumor agnostic accelerated approval for an SRI [18,19].

To support the regular approval, efficacy was demonstrated in 316 patients with locally advanced or metastatic *RET* fusion–positive NSCLC. Among 69 treatment-naïve patients, ORR was 84% (95% CI: 73–92) with a DOR of 20.2 months (95% CI: 13, not estimable). Among 247 patients previously treated with platinum-based chemotherapy, ORR was 61% (95% CI: 55–67) with a DOR of 28.6 months (95% CI: 20, not estimable). The most common adverse reactions (≥25%) in patients were edema, diarrhea, fatigue, dry mouth, hypertension, abdominal pain, constipation, rash, nausea, and headache.

For the tumor agnostic accelerated approval, efficacy was shown in 41 patients with *RET* fusion–positive tumors (other than NSCLC and thyroid cancer) with disease progression on or following prior systemic treatment or who had no satisfactory alternative treatment options. Patients received selpercatinib until disease progression or unacceptable toxicity. Among 41 evaluable patients, ORR was 44% (95% CI: 28–60) with a DOR of 24.5 months (95% CI: 9.2, not estimable). Tumor types with responses included pancreatic adenocarcinoma, colorectal, salivary, unknown primary, breast, soft tissue sarcoma, bronchial carcinoid, ovarian, small intestine, and cholangiocarcinoma. Confirmatory data is being generated (NCT03157128; LIBRETTO-001)).

On 4 August 2023, Lilly announced topline results from the LIBRETTO-431 study that compared selpercatinib with platinum-based chemotherapy plus pemetrexed, +/− pembrolizumab, as first line treatment for patients *RET* fusion–positive advanced NSCLC. These data have now been fully published and presented [20]. In this study 261 *RET*-fusion–positive NSCLC were randomly assigned to either selpercatinib or platinum-based chemotherapy +/− pembrolizumab. After a median follow-up time of 19 months the progression free survival (PFS) was 24.8 months for selpercatinib (95% CI: 16.9, to not reached) versus 11.2 months in the chemotherapy ++/− pembrolizumab intention-to-treat population (95% CI: 8.8–16.8), respectively [20].

In parallel, data from the LIBRETTO-531 Phase III study (NCT04211337), evaluating selpercatinib versus physician’s choice of the MKIs cabozantinib or vandetanib as an initial treatment for patients with advanced or metastatic *RET*-mutant MTC, have been published [21]. The study met its primary endpoint, demonstrating a statistically significant improvement in PFS. After a median follow-up time of 12 months, the median PFS for selpercatinib has not yet been reached and was 16.8 months (95% CI: 12.2–25.1) in the comparator arm. PFS at 12-months was 86.6% (95% CI: 79.1–91.6) in the selpercatinib arm and 65.7% (95% CI: 51.9–76.4) in the comparator arm, respectively. Adverse events observed with selpercatinib were generally consistent with those identified across the previously reported selpercatinib development program.

Selpercatinib continues to be studied in several clinical trials including in the adjuvant setting in LIBRETTO-432 (NCT04819100), a randomized Phase 3 trial comparing selpercatinib with placebo in Stage IB-IIIA *RET* fusion–positive in NSCLC patients previously treated with definitive surgery or radiation and in LIBRETTO-121 (NCT03899792), a phase 1/2 study in 50 pediatric patients with advanced *RET*-altered solid or primary central nervous system tumors.

Pralsetinib (BLU-667), developed by Blueprint Medicines, Cambridge, MA, USA, is also an SRI [22]. Pralsetinib also received accelerated approval from the FDA (4 September 2020) for the treatment of metastatic *RET* fusion–positive NSCLC and was granted conditional marketing authorization by the EMA (18 November 2021) [22].

In the ARROW open-label phase I/II trial, patients received pralsetinib orally at doses of 30–600 mg once daily during initial dose ranging. In the phase 2 dose-expansion part, patients initiated pralsetinib at the recommended phase 2 dose of 400 mg once daily continuously in 28-day cycles. In the first 233 patients enrolled with *RET* fusion–positive NSCLC, there were 87 previously treated patients and 27 treatment-naive patients with baseline measurable disease. The ORR was 61% (95% CI: 50–71) for patients with previous platinum-based chemotherapy, including five (6%) patients with a complete response and 70% (95% CI: 50–86) for treatment-naïve patients including three (11%) with a complete response. In this group of 233 patients with *RET* fusion–positive NSCLC, common treatment related adverse events (TRAE) of grade 3 or higher were neutropenia, (18%), hypertension (11%), and anemia (10%); there were no treatment-related deaths in this population [22].

The ARROW trial also included a cohort of *RET*-mutant MTC and *RET* fusion–positive thyroid cancers. 122 patients with *RET*-mutant MTC and 20 patients with *RET* fusion–positive thyroid cancers were enrolled and 79 patients were evaluable for efficacy. Among patients with baseline measurable disease, overall response rates were 71% (95% CI: 48–89) in patients with treatment-naive *RET*-mutant MTC and 60% (95% CI: 46–73) in patients who had previously received cabozantinib or vandetanib, or both, and 89% (95% CI: 52–100) in patients with *RET* fusion–positive thyroid cancer [7].

Common (≥10%) grade 3 and above TRAEs with *RET*-altered thyroid cancer were hypertension (17% of 142 patients), neutropenia (13%), lymphopenia (12%), and anemia (10%). Serious TRAEs were reported in 21 patients (15%), the most frequent (≥2%) of which was pneumonitis (five patients (4%)). Five patients (4%) discontinued due to TRAEs. One (1%) patient died due to a TRAE; this patient was diagnosed with interstitial pneumonitis on day 44 and later discontinued pralsetinib after two cycles owing to treatment-related *Pneumocystis jirovecii* pneumonia. Grade 3 and above pneumonia of any cause occurred in 17 (12%) patients, with a median time to resolution of 1.3 weeks (95% CI: 0.9–1.7). Pralsetinib approved indications (accelerated approval) were subsequently expanded in December 2020 to the treatment of advanced or metastatic *RET*-mutant MTC (in patients who require systemic therapy) and advanced or metastatic *RET* fusion–positive thyroid cancer (in patients who require systemic therapy and are radioactive iodine-refractory (if radioactive iodine is appropriate)) [7,23].

More recently, updated data for the first 281 patients with *RET* fusion–positive NSCLC to receive pralsetinib were published [24]. The ORR was 72% (95% CI: 60–82) for treatment-naïve patients and 59% (95% CI: 50–67) for patients with prior platinum-based chemotherapy. Median duration of response was not reached for treatment-naïve patients and 22.3 months for prior platinum-based chemotherapy patients. Tumor shrinkage was observed in all treatment-naïve patients and in 97% of patients with prior platinum-based chemotherapy. Median progression-free survival was 13.0 and 16.5 months, respectively. In patients with measurable intracranial metastases, the intracranial response rate was 70% (95% CI: 35–93), and all had received prior systemic treatment.

In treatment-naïve patients with *RET* fusion–positive NSCLC who received pralsetinib (n = 116), the most common grade 3–4 TRAEs were neutropenia (18%), hypertension (10%), increased blood creatine phosphokinase (9%), and lymphopenia (9%). Overall, 7% (20/281) discontinued due to TRAEs.

Results from the confirmatory phase III AcceleRET Lung study (NCT04222972) of pralsetinib versus standard of care in the first-line setting are pending.

In June 2023, Genentech (Roche Group) announced the voluntary withdrawal of the US indication for *RET*-mutant MTC, which had previously been granted accelerated approval (December 2020), citing that it was not feasible to pursue pralsetinib in this indication for full approval [25,26]. Additionally, earlier in 2023, Genentech, Inc, San Francisco, CA, USA, had discontinued its collaboration agreement with Blueprint Medicines for pralsetinib [27].

### 2.6. The Optimal Profile of a Next-Generation SRI

While clinical trials indicate selpercatinib and pralsetinib are effective and have an acceptable safety profile, some patients exhibit low sensitivity or are unable to tolerate the side effects of these drugs [19,26]. Tolerability issues for selpercatinib that can lead to dose reductions or discontinuations include the following: hepatotoxicity (ALT/AST increases); interstitial lung disease/pneumonitis; hypertension; QTc interval prolongation; hemorrhagic events; hypersensitivity; tumor lysis syndrome; or hypothyroidism [18]. Tolerability issues for pralsetinib that can lead to dose reductions or discontinuations include interstitial lung disease/pneumonitis, hypertension, hepatotoxicity, hemorrhagic events, or tumor lysis syndrome [24].

Potential new toxicities have been reported as more real-world experience of the first-generation SRIs has been gathered. In December 2022, Prete et al. [28], reported that among ten patients at their site with advanced MTC that were participating in the LIBRETTO-201 expanded-access clinical trial, eight patients (three of whom had effusion at baseline) developed new effusions in the chest, pericardium, abdomen and/or pelvis. Symptoms were relieved by reducing the selpercatinib dose. The adverse reaction of chylothorax was added to the selpercatinib US Prescribing Information in 2022 [19] and is defined as a clinically relevant serious adverse reaction (<2%).

While pralsetinib and selpercatinib are effective in inhibiting RET gatekeeper mutations (e.g., RET^V804L/M^) that are associated with resistance to MKIs, acquired solvent-front resistance mutations (e.g., RET^G810R/S/C^) have been identified after treatment with both agents [29,30,31,32]. The emergence of these on-target resistance mutations can lead to tumor escape and progressive disease after prolonged treatment with first-generation SRIs.

Gatekeeper mutations are secondary resistance mutations within conserved residues that control access to a hydrophobic sub-pocket of the RET kinase domain where drugs bind [33]. Mutation of residues such as V804 to leucine or methionine, which are bulky, results in modified ATP affinity and restricted drug binding, causing drug resistance, particularly to multi-kinase inhibitors. As these mutations are within the drug binding pocket, they are known as gatekeeper residues [34]. Drugs such as selpercatinib and pralsetinib are able to maintain inhibitory activity against *RET* V804 gatekeeper mutations; however, this activity is lost against solvent-front mutations. Mutations found within the RET kinase solvent front at position 810, through the substitution of glycine with bulky or charged residues, directly interfere with the binding of drugs such as selpercatinib. In contrast to gatekeeper mutations, mutations within the solvent front have minor effects on ATP affinity, suggesting that the loss of inhibitory activity is driven by direct interference of drug binding, as opposed to increased kinase activity [30].

Additionally, there is growing evidence that there are also off-target resistance mechanisms, such as amplification of *KRAS* and *MET* and co-mutations in *TP53* [35], which can occur in parallel or independently from acquired resistance mutations in the *RET* domain [31]. Combination of SRIs with other tyrosine kinase inhibitors (e.g., tepotinib, *MET* inhibitor) may therefore be required in order to prevent or address emergence of these non-RET-resistant pathways.

The ideal next-generation SRI should have broad activity across clinically important *RET* fusions and mutations including on-target acquired resistance mutations, preferably with improved safety and tolerability compared with the first-generation agents, to facilitate the evaluation of novel treatment combinations. It is also important for a next-generation SRI to have effective penetration of the CNS to suppress or address the emergence of brain metastases. The estimated lifetime risk of brain metastases is high among patients with *RET* fusion–positive NSCLC and prognosis is poor [36].

## 3. Next-Generation Selective RET Inhibitors in Clinical Development

Several novel next-generation SRIs are undergoing clinical development (Table 1). Here, we will discuss the next-generation SRIs that are currently in clinical development and their potential to meet the unmet clinical need for patients who progress on first generation SRIs.

### 3.1. BOS172738, Boston Pharmaceuticals

BOS172738 (zeteletinib, Boston Pharmaceuticals, Cambridge, MA, USA) is a potent, SRI with antitumor activity against a range of *RET* fusion proteins, as well as resistant *RET*-active site mutations with Kd values ≤ 1 nM for *RET*wt, *RET*(M918T), and *RET*(V804L/M) [37]. BOS172738 is being evaluated in a phase I clinical study (NCT03780517). The dose-escalation module has been completed, and results were presented at ASCO 2021 [38]. A total of 67 patients with *RET*-altered advanced solid tumors received once-daily oral doses of BOS172738 (10–150 mg). Intra-patient dose escalation and over-accrual to dose levels deemed to be safe were permitted. Study endpoints were safety, tolerability, and confirmed ORR. As of 11 December 2020 (dose-escalation module), BOS172738 exhibited a favorable safety profile, with most treatment-emergent adverse events (TEAEs) classified as grade ≤ 2 and deemed unrelated to the drug. The most common TEAEs were creatinine phosphokinase increase (54%), dyspnea (34%), facial edema, AST elevation, anemia (25% each), neutropenia, diarrhea (22% each), fatigue (21%), and constipation (20%). BOS172738 was not associated with hypertension or significant hepatoxicity. BOS172738 demonstrated broad anti-tumor activity with an ORR of 33% (n = 18/54), an NSCLC ORR of 33% (n = 10/30), and MTC ORR of 44% (n = 7/16, including one complete response), and one patient with *RET* fusion–positive pancreatic cancer reported a partial response. Responders included patients with brain metastases, with one patient whose brain lesion decreased by 43%. BOS172738 exhibited dose-dependent exposure, rapid absorption and an extended half-life of approximately 65 h. The study is completed, as per the last updated information on clinicaltrials.gov in October 2023 (accessed on 14 December 2023).

### 3.2. TPX0046, Turning Point Therapeutics

A study of TPX0046, an oral RET/SRC inhibitor in advanced solid tumors harboring oncogenic *RET* fusions or mutations, was terminated in May 2023 due to an adverse change in the risk–benefit profile (Turning Point Therapeutics, Inc., San Diego, CA, USA NCT04161391, accessed 3 November 2023) [39].

### 3.3. Vepafestinib (TAS0953/HM06), Helsinn Healthcare SA, Lugano, Switzerland

Vepafestinib was recently reported as having greater selectivity for RET than first-generation SRIs, with greater inhibitory activity against *RET*-WT and *RET*-V804 gatekeeper and *RETG810* solvent-front mutations in vitro [40]. Vepafestinib suppressed growth of allograft tumors harboring *RET* fusions and resistant mutations including *KIF5B–RET*G810R. Vepafestinib inhibited the growth of multiple lung cancer patient-derived cell lines harboring *RET* fusions with different N-terminal partners (*CCDC6*, *KIF5B*, *TRIM33*) and an *RET*C634W mutation–positive MTC cell line and was active in a number of NSCLC xenograft models. Preclinical models suggested that vepafestinib penetrates the CNS more than selpercatinib and is superior in decreasing CNS disease and extending survival in mice [41,42]. Vepafestinib is currently undergoing a biomarker-driven phase I/II clinical trial (MARGARET) in the US and Japan for patients with solid tumors driven by *RET* alterations that have progressed on existing therapies (NCT04683250, accessed 3 November 2023). At this point, no clinical data are available.

### 3.4. SY5007, Shouyao Holdings (Beijing) Co., Ltd., Beijing, China

SY5007 is undergoing investigation in a phase I/II trial in *RET*-altered advanced solid tumors in China (NCT05278364, accessed 3 November 2023). Data from this study were presented at ASCO in 2023 [43]. The trial enrolled previously treated patients with positive RET status. In the dose-escalation phase, patients were administered oral SY-5007 20 mg once daily or 20, 40, 80, 120, 160, 200 mg twice daily (BID) in 28-day cycles. Doses 160 and 200 mg twice daily were studied in a dose-expansion phase to determine a recommended phase II dose (RP2D). Three cohorts (*RET* fusion–positive NSCLC or thyroid cancers and *RET*-mutant MTC) at RP2D were expanded in eligible patients. Primary endpoints included safety, tolerability, maximum tolerated dose (MTD), and dose-limiting toxicities (DLTs). Secondary endpoints included PK and preliminary antitumor activity of SY-5007.

As of February 2023, 60 patients—55 *RET*-fusion NSCLC and 5 *RET*-mutant solid tumors—were enrolled into dose-escalation cohorts (n = 17) and dose-expansion cohorts (n = 43). In total, 55 (91.7%) patients experienced TRAEs, including increased AST (50.0%), increased ALT (41.7%) and diarrhea (41.7%). Grade ≥ 3 TRAEs were observed in 22 (36.7%) patients, including hypertension (15%), diarrhea (5%), and increased ALT (3.3%). PK studies showed SY-5007 was absorbed rapidly and exposure increased in a dose-dependent manner at doses between 20 and 160 mg BID and was saturated at 160 mg BID.

In 50 evaluable patients, the overall ORR and disease control rate (DCR) was 62.0% (95% CI: 47–75) and 94.0% (95% CI: 84–99), respectively. In summary, 29 patients (24 NSCLC, 4 MTC) received SY-5007 at 160 mg BID: 27 of 28 (96.4%) evaluable patients showed tumor regression, and ORR and DCR were 72.4% (95% CI: 52.3–87) and 89.7% (95% CI: 72.7–97.8), respectively. For NSCLC patients, the ORR and DCR were 75.0% (95% CI: 53–90) and 91.7% (95% CI: 73–99), respectively. A phase II clinical study of SY-5007 in NSCLC patients is ongoing.

### 3.5. EP0031—Kelun Biotech (KL590586/A400) and Ellipses Pharma (EP0031)

EP0031 is being developed jointly by Ellipses Pharma, London, UK, in the US, Europe, and other territories under the name EP0031 (NCT05443126) and Kelun Biotech, Chengdu, China, under the names KL590586 or A400, in China and other Asian countries (NCT05265091, accessed 3 November 2023).

Preclinical and clinical data from the Chinese study of EP0031 were presented at ASCO 2023 [44]. EP0031 was reported to have high selectivity for RET kinase compared to VEGFR2, greater potency than first-generation SRIs against common *RET* fusions and acquired on-target resistant mutations, including V804 gatekeeper and *RET* G810 solvent-front mutations, greater antitumor activity than first-generation SRIs in Ba/F3, a *KIF5B*-*RET*-G810R expressing cell line, and increased brain exposure and a higher brain/plasma ratio compared to first-generation SRIs in a mouse orthotopic brain tumor model, which correlated with improved survival.

The KL590586/A400 trial enrolled both treatment-naïve and previously treated patients with locally advanced or metastatic *RET*-altered solid tumors, including patients who had received prior SRIs. In the study, 109 patients received doses of 10–120 mg orally once daily during dose finding and expansion, the majority of whom had advanced *RET* fusion–positive NSCLC. Of those, 103 (94.5%) patients experienced any-grade TRAEs, with the majority being grade 1–2. The most common were increased ALT (51.4%), increased AST (48.6%) and constipation (31.2%). Grade 3 TRAEs were observed in 23.9% of patients, and included anemia (2.8%), increased ALT (1.8%) and increased AST (1.8%). A low frequency of dose interruptions (6.4%) and discontinuations (2.8%) was reported. PK studies showed dose linearity and steady state were achieved by the end of the cycle 1 [44].

ORR and DCR were 60% and 94%, respectively, in a total of 90 evaluable patients at multiple dose levels. For *RET* fusion–positive NSCLC patients (at 40–120 mg once daily (QD)), the ORR was 80.8% (95% CI: 60.7–93.5) and 69.7% (95% CI: 51.3–84.4) in treatment-naïve and previously treated patients, respectively. Responses appeared to be durable regardless of tumor type, *RET* fusion type, previous treatment or presence of brain metastases [44]. Five out of six patients with baseline RECIST-measurable CNS disease achieved an intracranial response. This study also showed tumor responses in NSCLC patients who were previously treated with first-generation SRIs. 

A phase I/II clinical study of A400/EP0031 at the recommended phase II dose level is ongoing, and additional cohorts for NSCLC have been initiated to support registration in China (NCT05265091).

In June 2022, Ellipses Pharma opened a parallel phase I/II study in the US (NCT05443126, accessed 3 November 2023) and Europe. Preliminary phase I dose-escalation data were presented at the AACR-NCI-EORTC International Conference on Molecular Targets and Cancer Therapeutics in October 2023 [45]. Data from a total of 15 patients—5 MTC, 2 CRC, and 8 NSCLC—were reported across 3 doses (20–90 mg QD). The PK profile was comparable with PK data from Chinese patients, and showed a dose-proportional increase in the dose level tested. No DLTs were reported and TEAEs were mainly grade 1 or 2. Preliminary efficacy data indicated activity in patients that were SRI-naïve to or had received a prior first-generation SRI.

### 3.6. HS-10365 Jiangsu Hansoh Pharmaceutical Co., Ltd., Lianyungang, China

A phase I study of HS-10365 has assessed the safety, tolerability, PK and anti-tumor activity in *RET*-altered solid tumors [46] (NCT05207787, accessed 3 November 2023), including *RET* fusion–positive NSCLC and *RET*-mutated MTC. Patients were dosed orally in 21-day cycles. This study is ongoing, and it is anticipated that approximately 273 participants with advanced solid tumors harboring an *RET* gene alteration will be enrolled to one of nine phase II cohorts.

As of December 2022 (as presented at AACR in 2023, [46]), 31 *RET* fusion–positive NSCLC patients received HS10365 at 6 doses (40 mg QD to 200 mg BID), including 25 who had previously received platinum-based chemotherapy and 6 treatment-naïve patients. Among all fusion variants, 15 patients had *KIF5B*, 14 patients had *CCDC6*, and 2 patients had other *RET*-fusion partners. Dose-limiting toxicity occurred only in one patient at 200 mg BID (grade 3 hypertension). The MTD was not determined, and 160 mg BID was the potential RP2D. The common (≥25%) TRAEs were AST increase, bilirubin increase, ALT increase, WBC decrease, platelet decrease, neutrophil decrease, serum creatinine increase, prolonged QT interval, hypoalbuminemia and anemia. Efficacy data were available for 30 *RET* fusion–positive NSCLC patients who had not received prior SRI: 24 pretreated and 6 treatment-naïve patients. The ORR was 70.0% (95% CI: 51–85), with 66.7% (16/24) in pretreated patients and 83.3% (5/6) in treatment-naïve patients. The DCR was 96.7% (95% CI: 83–100), with 95.8% (23/24) in pretreated patients and 100% (6/6) in treatment-naïve patients. Plasma exposure of HS-10365 increased proportionally following single and multiple doses. The mean plasma half-life of HS-10365 was approximately 5 to 9 h.

### 3.7. APS03118, Applied Pharmaceutical Science, Inc., Beijing, China

APS03118 is a next-generation SRI that is potent against a range of *RET* fusions and mutations, including both solvent-front and gatekeeper mutations. The selectivity, anti-RET activity, and intracranial efficacy of APS03118 were assessed in vitro and in vivo in a variety of RET-dependent tumor models [47].

APS03118 was highly selective against a panel of 468 kinases and demonstrated 130-fold selectivity over VEGFR-2. In enzymatic assays, APS03118 showed low nanomolar potency against wild-type *RET* and 25 *RET* mutations/fusions, including the inhibition of *RET*^G810R/C/S^ (IC50 0.04–5 nM) and *RET*^V804M/L/E^ (IC50 0.04–1 nM). APS03118 inhibited RET phosphorylation (IC50 < 15 nM) in Ba/F3 engineered RET cells (WT, G810R, V804M, M918T).

APS03118 demonstrated marked anti-tumor efficacy in vivo in RET-driven cell-derived (Ba/F3 *KIF5B-RET*, V804M, TT (C634W)) and patient-derived (*KIF5B-RET*, *CCDC6-RET*, *CCDC6-RET* V804M) xenograft tumor models at 10 mg/kg (TGI 87–108%). Tumors completely subsided in a *CCDC6-RET* orthotopic brain model with a 100% survival rate. In the Ba/F3 *KIF5B-RET* G810R xenograft model, APS03118 30 mg/kg showed 90% TGI and was well tolerated. APS03118 has received IND approval and fast-track designation from FDA, and a first-in-human phase I trial for patients with RET-driven solid tumors with activating *RET* alterations started in 2023 (currently recruiting in China), with an estimated enrollment of 35 patients (NCT05653869, accessed 3 November 2023).

### 3.8. LOXO-260, Loxo Oncology, Inc., Stamford, CT, USA, and Eli Lilly and Company, Indianapolis, IN, USA

LOXO-260 is an investigational anti-tumor drug targeting *RET* fusions and mutations, designed following the success of LOXO-292 (selpercatinib). LOXO-NGR-21001 (NCT05241834) is a global (currently recruiting in the US), open-label, first-in-human phase I study of LOXO-260 in patients with *RET* fusion–positive advanced solid tumors and *RET*-mutant MTC who received a prior SRI and other solid tumors with *RET* alterations refractory to SRI [48]. Phase Ia dose escalation will utilize a modified i3+3 design, allowing for patient backfill to previously cleared dose levels. Phase Ib dose expansion will evaluate LOXO-260 in specific expansion cohorts: *RET* fusion–positive NSCLC or thyroid cancers and *RET*-mutant MTC. No clinical data are currently available.

### 3.9. TY-1091, TYK Medicines, Inc.

TY-1091 (TYK Medicines, Inc., Shanghai, China) is another SRI scheduled for investigation in a phase I trial in China (NCT05675605, accessed 14 December 2023) that started recruiting in April 2023. The estimated completion date is June 2025. The planned trial is a dose-escalation study in advanced non-resectable NSCLC, advanced non-resectable thyroid cancer and other advanced solid tumors. A multicenter, open-label design is adopted for part 2 (expansion). One or two doses will be selected to conduct a dose-expansion trial in three cohorts.

### 3.10. HA121-28, CSPC ZhongQi Pharmaceutical Technology Co., Ltd.

Further phase I and II trials listed as recruiting in NSCLC and/or MTC include HA121-28 (CSPC ZhongQi Pharmaceutical Technology Co., Ltd. Shijiazhuang, China, NCT05117658 (phase II in NSCLC) and NCT04787328 (phase II in MTC), both accessed 3 November 2023); however, the last online updates posted were in 2022 for these studies.

### 3.11. HS269, Zhejiang Hisun Pharmaceutical Co., Ltd.

The phase I trial status of HS269, an SRI (Zhejiang Hisun Pharmaceutical Co., Ltd., Taizhou, China, NCT05058352, accessed 3 November 2023), is unknown (the study has passed its planned completion date and the status has not been verified in more than 2 years).

### 3.12. HEC169096, Sunshine Lake Pharma

HEC169096 (Sunshine Lake Pharma Co., Ltd., Dongguan, China, NCT05451602, accessed 3 November 2023) is listed as recruiting in NSCLC, MTC and other advanced solid tumors; however, the last online update posted was in 2022 for this study.

## 4. Discussion

Immense progress has been made in developing RET-targeting drugs, moving from unselective multi-kinase inhibitors (MKIs) to first-generation SRIs in a matter of a few years. In particular, the advent of the first-generation SRIs selpercatinib and pralsetinib resulted in meaningful benefits for patients in terms of deep and durable responses, which ultimately led to the accelerated approval of both drugs in patients with advanced lung and thyroid cancer in 2020. Most recently, in September 2022, selpercatinib was granted regular approval for adult patients with locally advanced or metastatic NSCLC with a *RET* fusion and was granted accelerated approval for *RET* fusion–positive cancers in a tumor agnostic setting [19]. Importantly confirmatory phase III studies in *RET* fusion–positive NSCLC and *RET*-mutated MTC have been recently reported, strengthening the data for selpercatinib in these settings [20,21].

Despite these advances, there is clear evidence that tumors can become less responsive over time and ultimately progress due to acquired resistance mutations. In this context, important discoveries have been made and “on- and off-target” escape mechanisms identified. Briefly, NSCLC or MTC patients who progressed after initial response to selpercatinib or pralsetinib revealed three emerging distinct resistance mutations, namely, G810R/S/C within the *RET* solvent front, which are associated with impaired binding of selpercatinib and pralsetinib to *RET* fusions. In addition, off-target *MET* or *KRAS* amplifications were detected in *RET* fusion–positive NSCLC patients who received either selpercatinib or pralsetinib [31,32,35].

At this point, several next-generation SRIs are in preclinical and clinical development aiming to address the emergence of resistance to the first-generation agents. The ideal profile of a next-generation agent has been described and includes broad activity against common *RET* fusions and mutations and resistant mutations, acceptable safety and tolerability, especially to support potential combination strategies, and must also have robust activity in the CNS.

As has been described, a number of agents are in development that seek to deliver this profile and clinical data are emerging. However, it is clear that further clinical data will be required before it can be confirmed that these candidates have the potential to follow and potentially surpass the first-generation agents in clinical practice. It is also clear that there is a need for evaluation of novel combination strategies with these agents with the aim of addressing resistance that is acquired through both *RET*-dependent and -independent pathways.

## 5. Conclusions

Activating *RET* mutations and rearrangements have been identified as actionable drivers of oncogenesis in numerous cancer types, including NSCLC and thyroid cancers. Current clinical data demonstrate that patients can achieve long, deep, and meaningful responses with first-generation SRIs. Their use in clinical practice is likely to evolve with potential future use as adjuvant treatment for *RET* fusion–positive NSCLC as well as expanded use across multiple solid tumors with *RET* mutations/fusions, as genomic testing becomes more widespread and eligible patients more readily identified. However, limitations of the currently approved SRIs include the inevitable development of acquired resistance and a safety and tolerability profile that is not optimal, especially for combination approaches. Currently, there are a number of companies developing next-generation SRIs with a focus on improving on the profile of first-generation agents, addressing resistance pathways, increasing activity in the CNS, and establishing superior tolerability. The potential of these agents has been demonstrated preclinically. Clinical trials are ongoing and promising data are beginning to emerge, with further data anticipated imminently.

## Figures and Tables

**Table 1 cancers-16-00031-t001:** Current clinical trials of next-generation selective RET inhibitors.

Drug Name Sponsor	Phase (Clinical Data Available) ^1^	Status	Trial Identifier
BOS172738 Boston Pharmaceuticals	Phase I (Phase I reported) [37,38]	Completed	NCT03780517
TPX0046 Turning Point	Phase I/II (Preliminary Phase I reported) [39]	Terminated	NCT04161391
Vepafestinib (TAS0953/HM06) Helsinn	Phase I/II [40,41,42]	Recruiting US, Japan	NCT04683250
SY5007 Shouyao Holdings	Phase I/II (Phase I reported) [43]	Recruiting China	NCT05278364
KL590586 (A400, EP0031) Kelun Biotech	Phase I/II (Phase I reported) [44]	Recruiting China	NCT05265091
EP0031 (A400, KL590586) Ellipses Pharma	Phase I/II (Preliminary Phase I reported) [45]	Recruiting US, Europe	NCT05443126
HS-10365 Jiangsu Hansoh Pharmaceutical	Phase I/II (Phase I reported) [46]	Recruiting China	NCT05207787
APS03118 Applied Pharmaceutical Science	Phase I [47]	Recruiting China	NCT05653869
LOXO-260 (LY3838915) Eli Lilly	Phase I [48]	Recruiting US	NCT05241834
TY-1091 TYK Medicines	Phase I/II	Recruiting China	NCT05675605
HA121-28 CSPC ZhongQi Pharmaceutical	Phase II: NSCLC Phase II: MTC	Recruiting China	NCT05117658 NCT04787328
HS269 Zhejiang Hisun Pharmaceutical	Phase I	Unknown China	NCT05058352
HEC169096 Sunshine Lake Pharma	Phase I/II	Recruiting China	NCT05451602

^1^ Clinical data reference is provided.

## Data Availability

No new data were created or analyzed in this review article. Data sharing is not applicable to this article.

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
