# Peer review of "Selective RET Inhibitors (SRIs) in Cancer: A Journey from Multi-Kinase Inhibitors to the Next Generation of SRIs"

_cancers, 2023, doi:10.3390/cancers16010031_

Round 1

Reviewer 1 Report

Comments and Suggestions for Authors

This is a timely review on the current state of therapeutics targeting the receptor tyrosine kinase RET. In particular, the authors discuss the trend from broad spectrum/mixed kinase inhibitors to the development of selective RET inhibitors (SRIs), the acquired resistance associated with this shift and how future drugs will meet this challenge. This paradigm has been seen with other cytoplasmic and receptor tyrosine kinases. However, this treatise brings a fresh perspective and details the initial success of first generation SRIs selpercatinib and pralsetinib in treating NSCLC and medullary thyroid cancer and the anticipated potential for clinical success of future second-generation  generation SRIs. 

Minor issues were noted, including the lack of defining abbreviations, such as ORR and the misspelling of Eli Lilly. Given that this manuscript will likely be of interest to both clinicians and bench scientists, defining gatekeeper mutations vs. solvent front mutations and their targeting by first and second-generation inhibitors would be helpful. A figure showing where these sites exist in the 3D structure of RET should be considered.

Author Response

Reviewer 1 comment: 
Minor issues were noted, including the lack of defining abbreviations, such as ORR and the misspelling 
of Eli Lilly. Given that this manuscript will likely be of interest to both clinicians and bench scientists, defining 
gatekeeper mutations vs. solvent front mutations and their targeting by first and second-generation 
inhibitors would be helpful. A figure showing where these sites exist in the 3D structure of RET should be 
considered.
Actions:
1. Defining abbreviations and spelling corrections made
2. New paragraph added on page 6 (lines 282-294 – in clean version) describing gatekeeper mutations 
and solvent front mutations and their targeting by multi-kinase inhibitors and first-generation
selective RET inhibitors. Note that a figure showing where these sites exist has not been added since 
this is already well described in the literature

Reviewer 2 Report

Comments and Suggestions for Authors

Overall, this is the well written comprehensive review article on the current landscape and upcoming advances in the field of RET inhibitors for Cancer treatment. The work pertinent given the unpredictable and not very strong response from initial multi kinase inhibitors along with toxicities. Following that, the currently approved selective RET inhibitors have improved on the efficacy and reduced the toxicity but there is still development of resistance escape of the cancer from these agents especially when it pertains to metastasis to the central nervous system. This has led to further investigation and continuing basic as well as clinical research in the field of next generation selective RET inhibitors.

The others provide a comprehensive table of the next generation agents.

Suggestions and edits:

Another table summarizing the multi kinase as well as the first-generation selective RET inhibitors, their usual response in approved cancers, as well as common toxicities would be helpful for the readers to have a quick glance for that information rather than parsing through all of the text.

Author Response

Reviewer 2 comment:
Another table summarizing the multi kinase as well as the first-generation selective RET inhibitors, their 
usual response in approved cancers, as well as common toxicities would be helpful for the readers to have a 
quick glance for that information rather than parsing through all of the text.
Note that an additional table has not been added because the reader is directed to the approved label for 
each therapy, and all therapies are well documented in the literature.
